# Agricultural and Socioeconomic Factors Associated with Farmer Household Dietary Diversity in India: A Comparative Study of Visakhapatnam and Sonipat

**Sukhwinder Singh** [1,*] ![ID], **Nikhil Srinivasapura Venkateshmurthy** [1,2] ![ID], **Kerry Ann Brown** [3],
**Avinav Prasad Maddury** [2], **Rajesh Khatkar** [2], **Prashant Jarhyan** [2], **Dorairaj Prabhakaran** [1,2] **and Sailesh Mohan** [1,2] ![ID]

[1] Centre for Chronic Disease Control, New Delhi 110016, India
[2] Public Health Foundation of India, Gurgaon 122002, India
[3] College of Life & Environmental Sciences, University of Exeter, Exeter EX4 4PY, UK
[*] Correspondence: sukhwinder@ccdcindia.org; Tel.: +91-783-777-9895

**Abstract:** Using primary data from 479 farmer households, this study examined the associations between agricultural and socioeconomic factors and farmer household dietary diversity in Visakhapatnam and Sonipat. Cropping intensity was positively associated with farmers' household dietary diversity score (HDDS), suggesting that higher cropping intensity may expand the gross cropped area and improve food security among subsistence farmers. Distance to food markets was also significantly associated with farmer HDDS, which suggests that market integration with rural households can improve farmer HDDS in Visakhapatnam. In Sonipat, wealth index had a positive association with farmer HDDS, targeting the income pathway by improving farmer HDDS in this region. Considering the relative contribution of these factors, distance to food markets, cropping intensity, and crop diversity were the three most important factors affecting farmer HDDS in Visakhapatnam, whereas wealth index, cropping intensity, and distance to food markets emerged as the top three important factors contributing to farmer HDDS in Sonipat. Our study concludes that the associations between agricultural and socioeconomic factors and farmer HDDS are complex but context- and location-specific; therefore, considering the site- and context-specific circumstances, different connections to HDDS in India can be found to better support policy priorities on the ground.

**Keywords:** agriculture; nutrition; crop diversity; household dietary diversity score; India

## 1. Introduction

Malnutrition is a global burden with nearly 123 countries facing a "triple burden" of energy excess, micronutrient deficiencies, and obesity [1,2]. About 281 million people in South Asia do not have adequate calorie intake catering to their nutritional needs. In India, about 32, 19, and 32% of children are reportedly stunted, wasted, or underweight, respectively, whereas 23% of adults are overweight. Furthermore, 67% of children, 57% of women, and 25% of men in India are anemic [3]. Although the Indian government is trying to reduce malnutrition among the rural poor through the "National Food Security Act (NFSA)" which provides subsidized food grains to poorer sections of society [4], the impacts of such programs are yet to be investigated [5].

Many studies [6–13] have examined various agricultural and socioeconomic factors associated with farmer household dietary diversity which is one of the key indicators of household nutrition. Some of these studies [10,12–14] have concluded that agricultural biodiversity, one of the major agricultural factors, is significantly associated with farmer household and individual dietary diversity in countries with low and medium income, although the magnitude of these associations was small. Higher farm incomes, another major agricultural factor, can improve farmer household dietary diversity [15]. Considering the socioeconomic factors, farmer education had a significant impact on farmer household

dietary diversity in India [16]. Some studies in southern African countries [8,9] reported that market integration can also influence household dietary diversity.

Although many studies have examined the association between agricultural and socioeconomic factors and household dietary diversity score (HDDS) among farmer households, few studies suggested that these drivers may vary across Indian states, which are completely different from each other in terms of agroclimatic conditions, agricultural practices, and socioeconomic setup. However, a recent study by Singh et al. [13] compared agricultural and socioeconomic factors associated with dietary diversity among farmers in two divergent states, Gujarat and Haryana, and found quite contrasting yet interesting results. Education and income were the two most important factors associated with dietary diversity in Gujarat while crop diversity, and market proximity influenced dietary diversity in Haryana. Further, Singh et al. [17] concluded that factors associated with farmer household dietary diversity were region-specific and vary across districts within each state. Therefore, for a deeper investigation of how the drivers of rural household nutrition vary across divergent Indian states, this study investigates the association between agricultural and socioeconomic factors and farmer HDDS across two districts, Visakhapatnam in Andhra Pradesh and Sonipat in Haryana, which have different agroclimatic conditions, socioeconomic setups, crop production, and food consumption patterns. Visakhapatnam is a large and densely populated coastal district of Andhra Pradesh, which is a rapidly modernizing urban site with a high dependency on local markets; on the other hand, Sonipat is a small town in Haryana where food systems in both urban and rural areas are less localized due to its proximity to New Delhi, the national capital of India (www.crida.in, accessed on 20 July 2020; www.agricoop.nic.in, accessed on 20 July 2020). This study aims to understand 'how agricultural and socioeconomic factors are associated with household dietary diversity, and how these associations vary across these two agroclimatically, socially, economically, and culturally divergent regions of India.' Specifically, this study investigates the following research questions:

1.  How are agricultural (cropping intensity, crop diversity) and socioeconomic factors (income diversity, family education, kitchen garden, milk production, distance traveled to markets, and wealth index) associated with farmer HDDS (overall, monsoon, winter, and summer) in Visakhapatnam and Sonipat?
2.  Which agricultural and socioeconomic factors have the largest associations with overall, monsoon, winter, and summer HDDS in Visakhapatnam and Sonipat?

This cross-sectional study has important implications for the current scenarios related to 'how crop specialization, particularly in Haryana, and wider farmer livelihood portfolios, particularly in Visakhapatnam, among farmer households may affect HDDS among farmer households.' These changes, given the fact that India has the highest malnutrition rates worldwide, are becoming more prominent across many Indian states as farming communities have become or are becoming better integrated with markets [18]. More recently, Bhagowalia et al. [15] and Kavitha et al. [19] suggested that crop diversity and household dietary diversity in India are positively associated. Therefore, the results from this study can help identify potential strategies and food policy interventions to improve dietary diversity in critically malnourished regions of India. From a policy perspective, the results of this study may help policymakers understand the drivers of farmer household HDDS in rural India, allowing them to devise tailor-made policies and targeted programs that can improve farmer household nutrition, particularly in Andhra Pradesh and Haryana.

## 2. Material and Methods

### 2.1. Study Locations and Sampling Methodology

For this study, quantitative data from 479 farmer households (Figure 1) in Visakhapatnam (Andhra Pradesh) and Sonipat (Haryana) were collected. They were selected purposively as they make an ideal case for a comparative study. Considering their agroclimatic conditions, 80% of soils in Visakhapatnam are either red clay loam or sandy loam soils; in Sonipat, all farms represent sandy loam soils. Only 36% of the gross cropped area

is irrigated in Visakhapatnam as compared to 98% in Sonipat (www.crida.in, accessed on 25 July 2020; www.agricoop.nic.in, accessed on 25 July 2020).

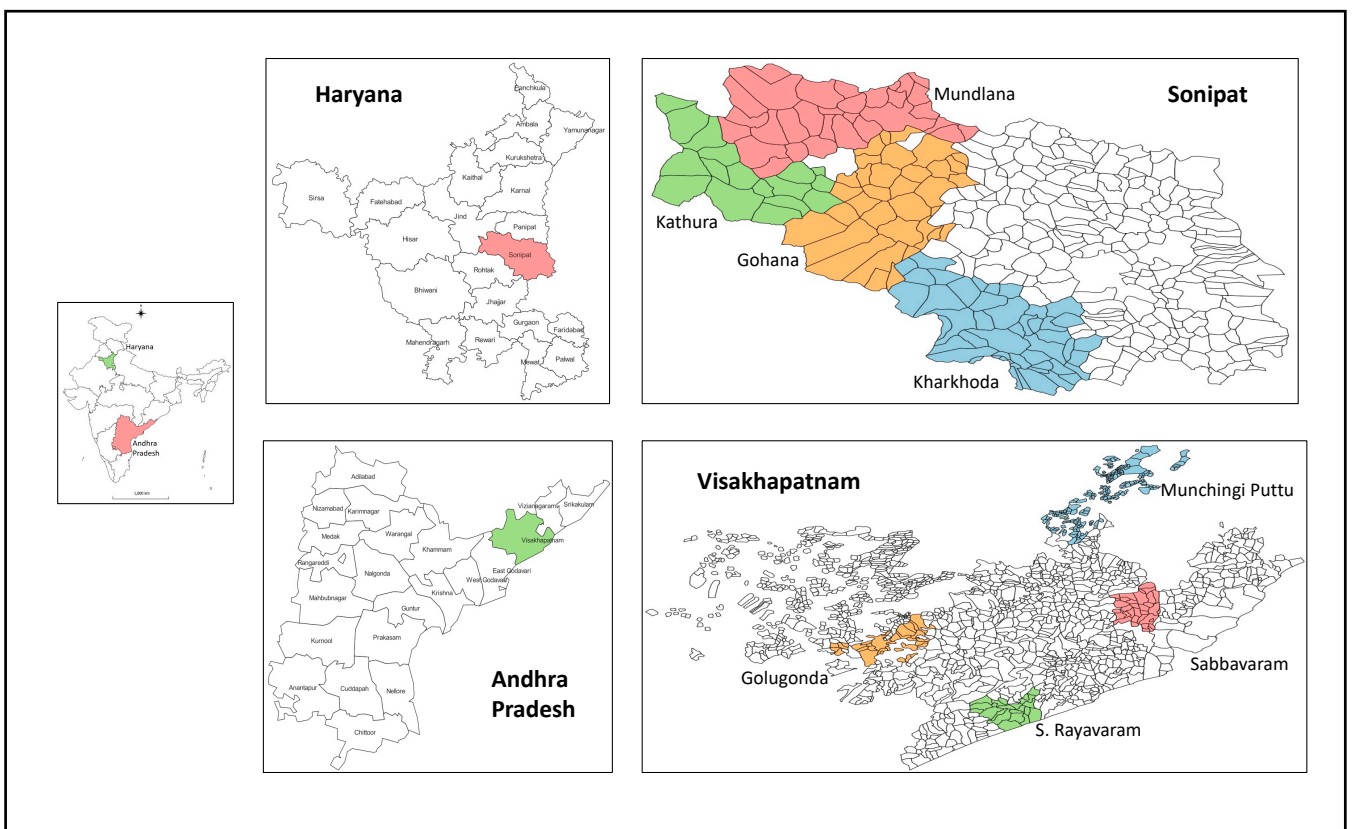

**Figure 1.** A map of states, districts, and blocks selected for this study.

Further, Visakhapatnam and Sonipat have contrasting crop production patterns. For instance, cropping patterns in Visakhapatnam are less intensive (129%) but quite diversified, with a range of crops from cereals such as rice, millets, e.g., finger millet (ragi), pearl millet (bajra), and sorghum (jowar) to cash crops such as sugarcane, groundnut, sesame, niger, and chillies; whereas Sonipat has very intensive (182%) but less diversified cropping patterns, largely concentrating on wheat and rice. (www.crida.in, accessed on 30 July 2020; www.agricoop.nic.in, accessed on 30 July 2020). To select blocks, villages, and then farmers within Visakhapatnam and Sonipat, we used a multistage cluster sampling technique. Using a farmer intensity index (FII adapted from [20,21]), we sampled four variant blocks (Mandals) within Visakhapatnam and Sonipat. FII was calculated by normalizing the indices of four agricultural factors, e.g., cropping intensity, crop diversity, groundwater development, and fertilizer input (Table 1). Then, a simple average of each of these four indices was used to realize the final FII for each of the four blocks. Based on the final FII indices, four blocks, one with the highest FII, two closest to average FII, and another with the lowest FII, were selected each in Visakhapatnam and Sonipat (Figure 1).

**Table 1.** Calculation of farming intensity index (FII) to select blocks/mandals in Visakhapatnam and Sonipat using major agricultural indicators.

| Sr. No | District | Cropping Intensity (%) | | Crop Diversity Index (CDI) | | Groundwater Development (%) | | Fertilizer Consumption (tons/ha) | | Farming Intensity Index (FII) |
|---|---|---|---|---|---|---|---|---|---|---|
| | Sonipat | X | A = (X − x̄)/SD | X | B=(X − x̄)/SD | X | C= (X − x̄)/SD | X | D = (X − x̄)/SD | (A + B + C + D)/4 |
| 1 | Sonipat | 160 | −0.85 | 0.52 | −1.12 | 129 | 0.10 | 69.82 | 1.29 | −0.14 |
| 2 | Rai | 147 | −1.46 | 0.49 | −1.74 | 196 | 1.78 | 63.91 | 0.52 | −0.23 |
| 3 | Kharkhoda | 173 | −0.22 | 0.60 | 0.99 | 143 | 0.45 | 62.60 | 0.35 | 0.39 |
| 4 | Gohana | 178 | 0.00 | 0.59 | 0.69 | 135 | 0.25 | 52.18 | −1.01 | −0.02 |
| 5 | Kathura | 180 | 0.11 | 0.59 | 0.53 | 78 | −1.18 | 49.45 | −1.37 | −0.48 |
| 6 | Mundlana | 208 | 1.42 | 0.61 | 1.10 | 80 | −1.13 | 55.21 | −0.61 | 0.20 |
| 7 | Ganaur | 199 | 1.00 | 0.55 | −0.45 | 120 | −0.13 | 66.29 | 0.83 | 0.31 |
| | District average (x̄) | 178 | | 0.56 | | 126 | | 59.93 | | |
| | Standard Deviation (SD) | 21 | | 0.04 | | 40 | | 7.67 | | |
| | Visakhapatnam | | | | | | | | | |
| 1 | Munchingi Puttu | 107 | −1.41 | 0.37 | −0.96 | 5 | −1.37 | 15 | −1.38 | −1.28 |
| 2 | Peda Bayalu | 111 | −0.97 | 0.42 | −0.54 | 3 | −1.48 | 17 | −1.36 | −1.09 |
| 3 | Dumbriguda | 125 | 0.38 | 0.35 | −1.10 | 6 | −1.32 | 12 | −1.39 | −0.86 |
| 4 | Araku Valley | 118 | −0.34 | 0.44 | −0.35 | 5 | −1.37 | 16 | −1.37 | −0.86 |
| 5 | Ananthagiri | 117 | −0.43 | 0.37 | −0.89 | 20 | −0.53 | 23 | −1.33 | −0.80 |
| 6 | Hukumpeta | 110 | −1.08 | 0.49 | 0.01 | 8 | −1.20 | 9 | −1.41 | −0.92 |
| 7 | Paderu | 113 | −0.80 | 0.44 | −0.38 | 11 | −1.04 | 13 | −1.39 | −0.90 |
| 8 | G.Madugula | 115 | −0.64 | 0.49 | 0.02 | 12 | −0.98 | 10 | −1.40 | −0.75 |
| 9 | Chintapalle | 132 | 1.00 | 0.51 | 0.19 | 14 | −0.87 | 10 | −1.40 | −0.27 |
| 10 | Gudem Kothaveedhi | 144 | 2.14 | 0.61 | 0.96 | 6 | −1.32 | 11 | −1.40 | 0.09 |
| 11 | Koyyuru | 106 | −1.44 | 0.38 | −0.86 | 12 | −0.98 | 43 | −1.21 | −1.12 |
| 12 | Nathavaram | 126 | 0.41 | 0.78 | 2.27 | 19 | −0.59 | 337 | 0.51 | 0.65 |
| 13 | Golugonda | 117 | −0.39 | 0.48 | −0.07 | 20 | −0.53 | 386 | 0.80 | −0.05 |
| 14 | Narsipatnam | 116 | −0.55 | 0.63 | 1.11 | 29 | −0.03 | 372 | 0.71 | 0.31 |
| 15 | Rolugunta | 117 | −0.43 | 0.31 | −1.37 | 44 | 0.81 | 376 | 0.74 | −0.06 |
| 16 | Ravikamatham | 112 | −0.87 | 0.42 | −0.54 | 30 | 0.02 | 408 | 0.93 | −0.11 |
| 17 | Madugula | 134 | 1.17 | 0.48 | −0.11 | 12 | −0.98 | 395 | 0.85 | 0.23 |
| 18 | Cheedikada | 134 | 1.22 | 0.39 | −0.78 | 32 | 0.14 | 400 | 0.88 | 0.36 |
| 19 | Devarapalle | 129 | 0.70 | 0.44 | −0.40 | 34 | 0.25 | 401 | 0.88 | 0.36 |
| 20 | K.Kotapadu | 124 | 0.29 | 0.38 | −0.88 | 50 | 1.14 | 371 | 0.71 | 0.32 |
| 21 | Sabbavaram | 130 | 0.86 | 0.34 | −1.12 | 31 | 0.08 | 303 | 0.31 | 0.03 |
| 22 | Pendurthi | 122 | 0.10 | 0.57 | 0.61 | 27 | −0.14 | 355 | 0.62 | 0.30 |
| 23 | Anandapuram | 122 | 0.04 | 0.51 | 0.18 | 36 | 0.36 | 230 | −0.12 | 0.12 |
| 24 | Padmanabham | 138 | 1.64 | 0.46 | −0.19 | 43 | 0.75 | 321 | 0.42 | 0.65 |
| 25 | Bheemunipatnam | 139 | 1.71 | 0.44 | −0.37 | 36 | 0.36 | 389 | 0.81 | 0.63 |
| 26 | Visakhapatnam (R) | 109 | −1.23 | 1.00 | 3.92 | 23 | −0.37 | 5 | −1.43 | 0.22 |
| 27 | Visakhapatnam (U) | 0 | 0 | 0 | 0 | 0 | 0 | 0 | 0 | 0 |
| 28 | Pedagantyada | 101 | −1.96 | 0.42 | −0.52 | 7 | −1.26 | 126 | −0.73 | −1.12 |
| 29 | Gajuwaka | 138 | 1.55 | 0.55 | 0.43 | 12 | −0.98 | 0.69 | −1.46 | −0.12 |
| 30 | Paravada | 109 | −1.23 | 0.47 | −0.14 | 33 | 0.19 | 138 | −0.66 | −0.46 |
| 31 | Anakapalle | 124 | 0.30 | 0.41 | −0.61 | 52 | 1.25 | 405 | 0.91 | 0.46 |
| 32 | Chodavaram | 118 | −0.29 | 0.40 | −0.68 | 55 | 1.42 | 397 | 0.86 | 0.33 |
| 33 | Butchayyapeta | 127 | 0.55 | 0.40 | −0.66 | 25 | −0.26 | 396 | 0.86 | 0.12 |
| 34 | Kotauratla | 110 | −1.08 | 0.47 | −0.14 | 41 | 0.64 | 376 | 0.74 | 0.04 |
| 35 | Makavarapalem | 110 | −1.14 | 0.47 | −0.17 | 46 | 0.92 | 373 | 0.72 | 0.08 |
| 36 | Kasimkota | 123 | 0.11 | 0.53 | 0.30 | 32 | 0.14 | 395 | 0.85 | 0.35 |
| 37 | Munagapaka | 130 | 0.85 | 0.41 | −0.60 | 49 | 1.08 | 365 | 0.67 | 0.50 |
| 38 | Atchutapuram | 130 | 0.84 | 0.33 | −1.20 | 47 | 0.97 | 359 | 0.64 | 0.31 |
| 39 | Yelamanchili | 128 | 0.66 | 0.49 | 0.02 | 54 | 1.36 | 413 | 0.95 | 0.75 |
| 40 | Nakkapalle | 107 | −1.38 | 0.65 | 1.21 | 42 | 0.69 | 364 | 0.67 | 0.30 |
| 41 | Payakaraopeta | 122 | 0.04 | 0.72 | 1.77 | 69 | 2.20 | 405 | 0.91 | 1.23 |
| 42 | S.Rayavaram | 128 | 0.64 | 0.69 | 1.53 | 65 | 1.98 | 411 | 0.94 | 1.27 |
| 43 | Rambilli | 126 | 0.45 | 0.59 | 0.78 | 45 | 0.86 | 346 | 0.56 | 0.66 |
| | District average (x̄) | 121 | | 0.49 | | 29.57 | | 249.95 | | |
| | Standard Deviation (SD) | 10.43 | | 0.13 | | 17.91 | | 170.80 | | |

Using the same methodology, three divergent villages were selected within each of the selected blocks. However, due to non-availability of secondary data related to these variables at the village level, local extension workers were consulted to select villages in each block. The geographical location of blocks within districts and villages within blocks was also considered to select a set of geographically divergent blocks and villages within each district and block, respectively.

Within each of the selected villages, using a stratified random technique, approximately 20 farmer households belonging to different social groups (SC/ST, OBC, and General) and cultivating small, medium, and large landholdings were selected so that the sample drawn from each village represents all the social and landholder groups of the village concerned. In total, 479 farmers spread across 24 villages and 8 blocks in both districts were surveyed (Figure 2).

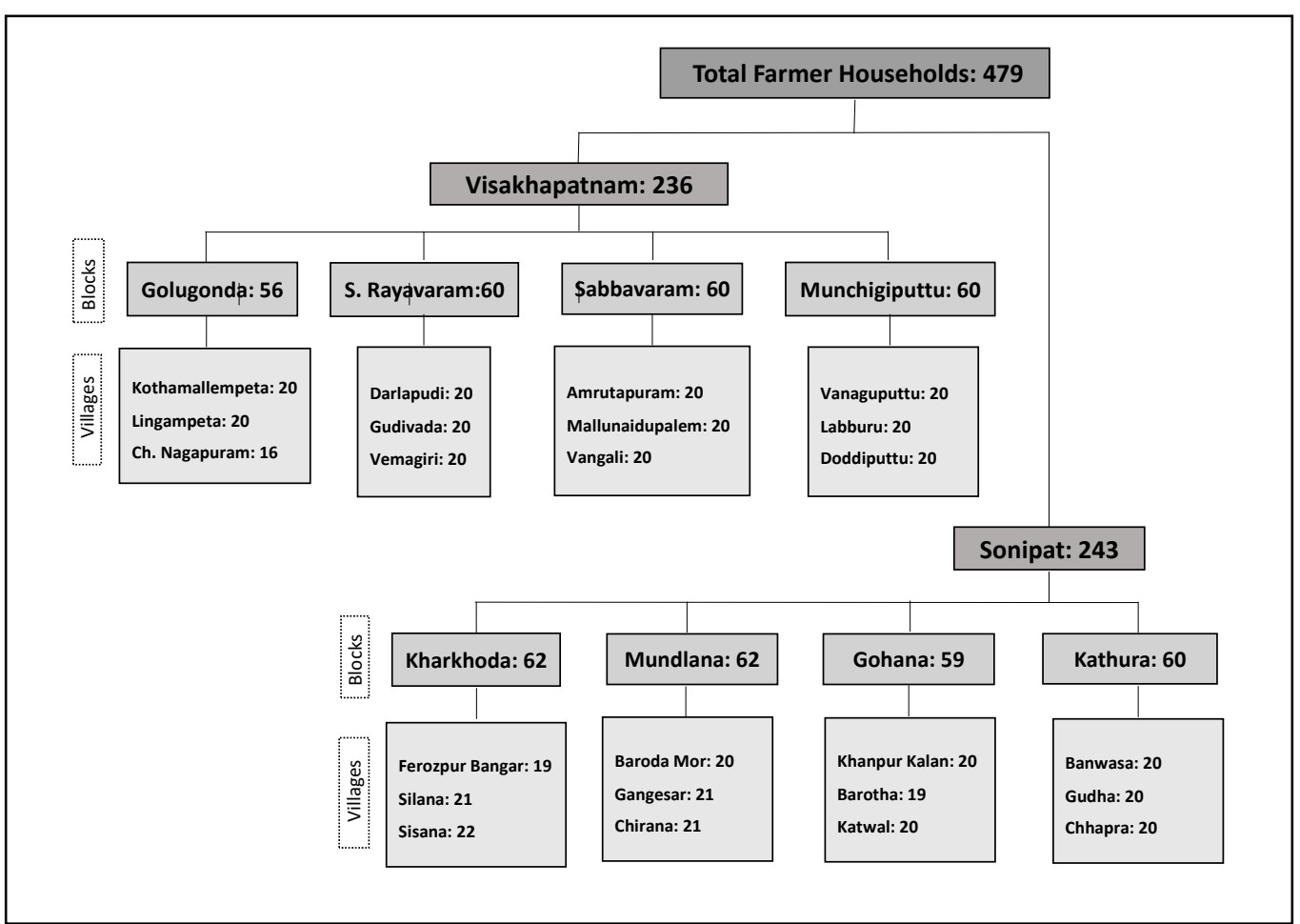

**Figure 2.** Sample distribution showing the number of farmer households surveyed in each village and block in Visakhapatnam and Sonipat.

## 2.2. Data Collection

Using Survey CTO, a mobile-based application, quantitative data related to cropping patterns, food production systems, household income sources, farmer household dietary patterns, and sources of food items in three seasons (monsoon, winter, and spring) were collected. Survey interviews were conducted at farmers' homes/farms, and village community centers within each village. This survey was reviewed and approved by the CCDC Institution Ethics Committee (IRB Approval Number: IRB00006330).

## 2.3. Metrics Constructed

Household Dietary Diversity Score: The Household Dietary Diversity Score (HDDS) is used to measure food availability and food accessibility aspects of food security. It makes an assessment of the number of different food groups consumed in the household during a defined reference period [9,22–24]. Using the FANTA 2006 guidelines [25], from each farmer household, we collected data related to 12 food groups (e.g., 1. Cereals; 2. Root and tubers; 3. Vegetables; 4. Fruits; 5. Meat, poultry, offal; 6. Eggs; 7. Fish and seafood; 8. Pulses/legumes/nuts; 9. Milk and milk products; 10. Oil/fats; 11. Sugar/honey; and 12. Miscellaneous) to calculate farmer HDDS across three seasons, namely, monsoon, winter, and summer. The total number of food groups consumed by a farmer household were tabulated as 0 (No) or 1 (Yes). The HDDS is a continuous score between 0 and 12 for each farmer household. Using the devised formula in the FANTA 2006 guidelines, we calculated HDDS for each farmer household by adding all the food items consumed (Sum = 1 + 2 + 3 + 4 + 5 + 6 + 7 + 8 + 9 + 10 + 11 + 12) in each season. For instance, if a

farmer household reported consuming 10 of 12 food groups in monsoon, the monsoon HDDS for that household was 10. The annual HDDS was calculated by averaging all three seasonal HDDSs.

Cropping Intensity: Cropping intensity (CI) is the number of times a crop is planted in a given agricultural area each year [26] (Siebert et al., 2010). Quantitatively, it is the ratio of the effective crop area harvested (gross cropped area) to a physical area (net sown area) [27]. Thus, we calculated cropping intensity for each farmer household using the gross cropped area and net sown area (Gross cropped area/net sown area*100).

Crop Diversification Index: Using the methodology used by Singh and Benbi [21], Singh et al. [13], and Singh et al. [17], we calculated the crop diversification index (CDI) for all farmer households. Using the 1-H formula, H (Hirschman–Herfindahl index—HHI) is measured as:

$$H = \sum_{i=1}^{N} S_i^2$$

where,

$N$ is the total number of crops during 2019–2020;

$S_i$ represents the area proportion of the i-th crop in the total cropped area.

$H$ equals 1 in the case of monoculture, and it approaches zero with an increasing level of crop diversity. When using 1-H, a larger number indicates higher crop diversity. It was calculated using all crops grown during the entire agricultural year (2019–2020).

Family Education Index: The family education index (FEI) for each farmer household was realized by adding the educational level of all adult and adolescent members in the farmer household concerned and then dividing the resulting value by the total number of adults and adolescents in that household. We considered the average level of education of all adults and adolescents rather than the maximum level of education within a given household because decisions related to household diets are mostly influenced by the backgrounds of multiple family members and not just the most educated member of the household concerned.

Income Diversity Index: Using the Hirschman–Herfindahl index, the income diversity Index (IDI) was calculated using the percentage of family income from different farm and non-farm sources, e.g., crop production, non-farm activities e.g., dairy, poultry, beekeeping, etc., business, government, or private employment. The farmer households with the most diversified income portfolio had the largest IDI. We asked farmers the estimated percentage of income coming from each source they mentioned rather than the total income from each of the sources because most farmers do not maintain income and expenditure accounts.

Wealth Index: The wealth index, which is a composite measure of a household's cumulative living standard, is calculated using data related to farmer households' ownership of selected assets, e.g., televisions, means of transport, bicycles, housing construction material, and drinking water and sanitation facilities. Following the procedure laid down by the DHS wealth index [28], we calculated the wealth index for each farmer household using the data collected on 26 household items used by the National Family Household Surveys in India [3].

### 2.4. Framework to Examine Associations

Several regressions were run to examine the associations between various socioeconomic and agricultural factors (as independent variables) and farmer HDDS (as a dependent variable) across three seasons—monsoon, winter, summer, and overall—separately for Visakhapatnam and Sonipat. For a robustness check, all regressions were run with block-fixed effects (Figure 3).

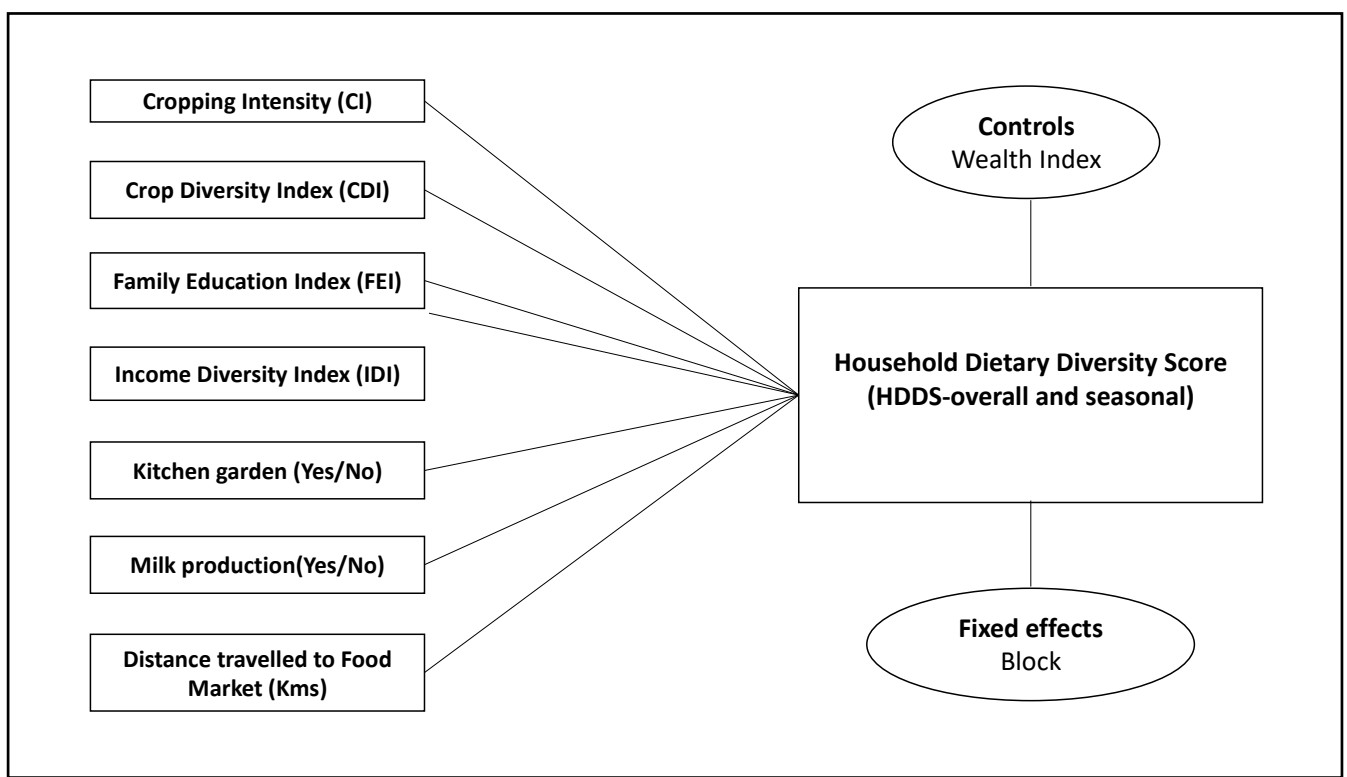

**Figure 3.** A conceptual framework outlining the independent variables, controls, and fixed effects used in regressions.

### 2.5. Statistical Modeling

To understand the locational variations in the basic variables related to agricultural and socioeconomic factors, we first tabulated their descriptive statistics for farmer households in Visakhapatnam and Sonipat. We then ran a series of regressions to identify the associations between study variables. For all analyses, linear regressions were run using R Project Software. All continuous independent variables were normalized to make coefficient values comparable across all independent variables. Upon calculating the variance inflation factors (VIFs) for each regression, we did not find any evidence of multi-collinearity (VIF < 1.2). Further, to minimize the effect of location on our regression results, we included block-fixed effects. The relaimpo package in R was used to assess the relative importance of independent variables used in each regression.

## 3. Results

### 3.1. Descriptive Statistics

Socioeconomic profile: Considering the educational status of farmer households surveyed at both study locations (Table 2), per capita education was 7.1 years. It was significantly ($p < 0.01$) higher in Sonipat (8.6 years; SD = 2.3) than in Visakhapatnam (5.5 years; SD = 3.1). An average farmer household in Visakhapatnam reported its per-capita annual income from all sources as USD 163 (Rs 12,918; SD = 10,663) while the respective figure for Sonipat was just USD 34 (Rs 2675; SD = 8482), which is 20% of the average figure reported by farmers in Visakhapatnam. However, on the contrary, wealth index (WI) for Sonipat farmers (9.2; SD = 4.6) was significantly higher ($p < 0.01$) than those in Visakhapatnam (−4.7; SD = 2.3). Farmer livelihood portfolios were more diverse in Visakhapatnam (0.4; SD = 0.2) than in Sonipat (0.18; SD = 0.22), which means more farmers in Visakhapatnam were into non-farming occupations, such as dairy or poultry farming, salaried jobs, and other non-farming business activities, compared to farmers in Sonipat who were largely relying on crop production.

**Table 2.** Socioeconomic, agricultural, and dietary profile of farmer households surveyed in Visakhapatnam (Andhra Pradesh) and Sonipat (Haryana).

| Parameters | Visakhapatnam (n = 236) | | Sonipat (n = 243) | | Overall (n = 479) | | Variation, if Significant (*t*-Test) |
|---|---|---|---|---|---|---|---|
| | Mean | SD | Mean | SD | Mean | SD | |
| Family education (years) | 5.5 | 3.1 | 8.6 | 2.3 | 7.1 | 3.1 | *p* < 0.01 |
| Per-capita annual income (Rs) | 12,918 | 10,663 | 2675 | 3301 | 8482 | 9709 | *p* < 0.01 |
| Wealth index (WI) | (−) 4.7 | 2.3 | (−) 9.2 | 4.6 | (−) 7.0 | 4.8 | *p* < 0.01 |
| Income diversity index (IDI) | 0.4 | 0.2 | 0.18 | 0.22 | 0.3 | 0.2 | *p* < 0.01 |
| Landholding size (ha) | 1.3 | 3.3 | 3.8 | 9.4 | 2.5 | 7.7 | *p* < 0.01 |
| Total cropped land area (ha) | 1.4 | 3.2 | 6.9 | 17.3 | 4.2 | 14.2 | *p* < 0.01 |
| Cropping intensity (%) | 124 | 46 | 186 | 35.1 | 156 | 51.32 | *p* < 0.01 |
| Crop diversity index (CDI) | 0.55 | 0.23 | 0.58 | 0.11 | 0.56 | 0.18 | Not significant |
| Number of crops in a year (#) | 3.4 | 1.83 | 3.3 | 1.19 | 3.4 | 1.54 | Not significant |
| Groundwater depth (feet) | 118 | 80 | 39 | 34 | 78 | 73.02 | *p* < 0.01 |
| Kitchen garden (Y/N) | 9 | Not applicable | 7 | Not applicable | 8 | Not applicable | Not applicable |
| Growing without fertilizers for domestic consumption (Y/N) | 31 | Not applicable | 7 | Not applicable | 19 | Not applicable | Not applicable |
| Growing without pesticides for domestic consumption (Y/N) | 37 | Not applicable | 8 | Not applicable | 22 | Not applicable | Not applicable |
| Started using less amount of fertilizer in last 5 years (Y/N) | 10 | Not applicable | 2 | Not applicable | 6 | Not applicable | Not applicable |
| Started using less amount of pesticide in last 5 years (Y/N) | 7 | Not applicable | 1 | Not applicable | 4 | Not applicable | Not applicable |
| Fruit and vegetables without fertilizers (Y/N) | 5 | Not applicable | 2 | Not applicable | 4 | Not applicable | Not applicable |
| Fruit and vegetables without pesticides (Y/N) | 4 | Not applicable | 2 | Not applicable | 3 | Not applicable | Not applicable |
| Milk produced (liters/daily) | 4.5 | 9.4 | 6.1 | 6.3 | 5.3 | 7.9 | Not significant |
| Number of poultry (currently) | 11.05 | 11.6 | Not applicable | Not applicable | 11.05 | 11.6 | Not applicable |
| Number of eggs (daily) | 4.3 | 5.9 | Not applicable | Not applicable | 4.3 | 5.7 | Not applicable |
| Distance traveled to buy food (km) | 4.9 | 4.9 | 4.7 | 4.3 | 4.8 | 4.6 | Not significant |
| Household dietary diversity score (overall) | 10.5 | 0.6 | 8.3 | 0.6 | 9.4 | 1.25 | *p* < 0.01 |
| Household dietary diversity score (monsoon) | 10.8 | 0.6 | 8.6 | 0.5 | 9.7 | 1.25 | *p* < 0.01 |
| Household dietary diversity score (winter) | 10.8 | 0.6 | 8.7 | 0.7 | 9.7 | 1.25 | *p* < 0.01 |
| Household dietary diversity score (summer) | 9.8 | 0.6 | 7.6 | 0.6 | 8.7 | 1.29 | *p* < 0.01 |

Agricultural profile: While looking at the average landholding size (Table 2), a typical farmer household was operating 2.5 hectares (ha) with 1.3 ha (SD = 3.3) in Visakhapatnam and 3.8 ha (SD = 9.4) in Sonipat, which were found to be statistically different (*p* < 0.01). Higher standard deviation figures suggest higher variability in landholding sizes within Visakhapatnam (Range = 0.2 ha to 16 ha) and Sonipat (Range = 0.16 ha to 21.2 ha). Considering the farmer distribution in terms of landholding size, 89% of the farmers in Visakhapatnam and 42% in Sonipat were smallholders cultivating less than 2 ha, suggesting that more than half of the farmers surveyed in Sonipat were medium and large landholders with relatively larger landholdings compared with those in Visakhapatnam. When looking at the cropping patterns, only 24% of the land area in Visakhapatnam was sown twice a given year (i.e., making the cropping intensity of 124%) while, in Sonipat, the cropping intensity was 186%, which means 86% of the land was sown twice in a given year. Further, cropping intensity varied significantly (*p* < 0.01) both in Visakhapatnam and Sonipat. On the other hand, cropping patterns at both the study locations were not very diverse, i.e., most farmers are into monocropping; that is, sowing 3.4 crops across two seasons in a given year. Further, crop diversity between the two locations was statistically comparable with low standard deviation figures, suggesting that crop diversity, even in the blocks and villages within Visakhapatnam and Sonipat, did not vary much.

As per farmers' estimates, groundwater depth was 118 feet (SD = 80) in Visakhapatnam whereas it was only 39 feet (SD = 34) in Sonipat, which varied significantly (*p* < 0.01).

Despite having access to land and water resources and human capital, only 8% of farmer households had a kitchen garden. While considering the health awareness among farmers, only 19 and 22% of farmers in both states, respectively, were growing crops for domestic consumption without using fertilizers and pesticides and such farmers were found more in Visakhapatnam (31%; 37%) than in Sonipat (7%; 8%). Similarly, very few farmers (4–6%) farmers reported starting using less fertilizers and pesticides in the last 5 years and only 3–4% of farmers were growing fruits and vegetables without using fertilizers and pesticides. Such farmers were found to be more in Visakhapatnam than in Sonipat. On average, a typical farmer household was producing 5.3 L (SD = 7.9) of milk daily, with 4.8 L (SD = 9.4) in Visakhapatnam and 6.1 L (SD = 6.3) in Sonipat. In Visakhapatnam, an average farmer household had 11 birds (SD = 11.6) in poultry and was producing 4.3 eggs (SD = 5.9) daily. However, no farmer household in Sonipat had poultry or was producing eggs. While looking at the market integration, farmers had to travel 4.8 km (SD = 4.6) with 4.9 km (SD = 4.9) in Visakhapatnam and 4.7 km (SD = 4.3) in Sonipat.

Household dietary diversity profile: Out of 12 food groups, an average farmer household in Visakhapatnam was consuming 10.5 food groups (SD = 0.6) compared with 8.3 food groups (SD = 0.6) in Sonipat and these figures varied significantly ($p < 0.01$). These trends were similar in the monsoon and winter seasons. However, in the summer season, the average figures were slightly lower at both study locations (Visakhapatnam = 9.8; Sonipat = 7.6). While looking at the consumption of individual groups (Table 3), cereals, root and tubers, vegetables, and fruits were consumed by almost all (99–100%) farmer households in all three seasons at both locations. Non-vegetarian food groups such as meat, poultry, offal, eggs, and fish were consumed by 97–99% of farmer households in Visakhapatnam across all three seasons, while none or very few farmer households in Sonipat consumed these food groups. Only 7% of farmer households in Sonipat reported consuming eggs during winter. All farmer households in Visakhapatnam were consuming pulses, legumes, and nuts across all three seasons, while the respective figures for Sonipat were around 88–89%. On the other hand, all farmer households were consuming milk and milk products across all seasons in Sonipat, whereas the respective figures for Visakhapatnam were around 94–95%. All farmer households at both locations reported consuming oil and fats and sugar and honey across all three seasons. Other food items (under the miscellaneous food category) were consumed by 69–70% of farmer households in Sonipat, while only 3% of farmer households in Visakhapatnam reported consuming these food items in winter and summer seasons.

**Table 3.** Food groups consumed by farmer households (%) during monsoon, winter, and summer and their source if consumed. Average distance (in km) farmers traveled to buy these food items.

| Food Group | Location | Consumed by the Number of Farmer Households (%) | | | If Consumed | |
| | | Monsoon | Winter | Summer | Produced (%) | Bought (%) |
|---|---|---|---|---|---|---|
| Cereals | Visakhapatnam | 100 | 99 | 100 | 84.32 | 53.39 |
| | Sonipat | 100 | 100 | 100 | 97.94 | 6.17 |
| | Overall | 100 | 100 | 100 | 91.23 | 29.44 |
| Root and tubers | Visakhapatnam | 99 | 99 | 99 | 5.93 | 96.61 |
| | Sonipat | 100 | 100 | 100 | 5.35 | 96.30 |
| | Overall | 99 | 100 | 99 | 5.64 | 96.45 |

**Table 3.** *Cont.*

| Food Group | Location | Consumed by the Number of Farmer Households (%) | | | If Consumed | |
| | | Monsoon | Winter | Summer | Produced (%) | Bought (%) |
|---|---|---|---|---|---|---|
| Vegetables | Visakhapatnam | 100 | 100 | 99 | 23.31 | 93.64 |
| | Sonipat | 100 | 100 | 100 | 7.82 | 95.47 |
| | Overall | 100 | 100 | 99 | 15.45 | 94.57 |
| Fruits | Visakhapatnam | 99 | 100 | 99 | 0.42 | 99.58 |
| | Sonipat | 100 | 100 | 100 | 1.65 | 98.77 |
| | Overall | 99 | 100 | 100 | 1.04 | 99.16 |
| Meat, poultry, and offal | Visakhapatnam | 97 | 97 | 97 | 7.20 | 94.49 |
| | Sonipat | 1 | 2 | 1 | 0.41 | 2.47 |
| | Overall | 48 | 49 | 48 | 3.76 | 47.81 |
| Eggs | Visakhapatnam | 97 | 97 | 97 | 3.81 | 95.76 |
| | Sonipat | 0 | 7 | 0 | 0.00 | 6.58 |
| | Overall | 48 | 51 | 48 | 1.88 | 50.52 |
| Fish and seafood | Visakhapatnam | 99 | 99 | 99 | 0.00 | 98.73 |
| | Sonipat | 0 | 0 | 0 | 0.00 | 0.00 |
| | Overall | 49 | 49 | 49 | 0.00 | 48.64 |
| Pulses/legumes/nuts | Visakhapatnam | 100 | 100 | 100 | 14.41 | 95.34 |
| | Sonipat | 88 | 89 | 88 | 2.06 | 87.24 |
| | Overall | 94 | 94 | 94 | 8.14 | 91.23 |
| Milk and milk products | Visakhapatnam | 90 | 90 | 90 | 42.80 | 54.24 |
| | Sonipat | 100 | 100 | 99 | 85.19 | 24.28 |
| | Overall | 95 | 95 | 94 | 64.30 | 39.04 |
| Oil and fats | Visakhapatnam | 100 | 100 | 100 | 0.42 | 100.00 |
| | Sonipat | 100 | 100 | 100 | 20.16 | 82.30 |
| | Overall | 100 | 100 | 100 | 10.44 | 91.02 |
| Sugar and honey | Visakhapatnam | 100 | 100 | 100 | 0.42 | 99.15 |
| | Sonipat | 100 | 100 | 100 | 0.82 | 99.18 |
| | Overall | 100 | 100 | 100 | 0.63 | 99.16 |
| Miscellaneous (condiments, tea, and coffee) | Visakhapatnam | 0 | 3 | 3 | 0.00 | 3.39 |
| | Sonipat | 69 | 70 | 69 | 0.41 | 69.14 |
| | Overall | 35 | 37 | 37 | 0.21 | 36.74 |

While looking at the food sources, cereals (91%) and milk (64%) were largely produced at home, while for the other food groups, most farmer households at both study locations relied on markets. However, there are some variations in some cases. For instance, 23% of farmer households in Visakhapatnam were producing vegetables while the respective figure for Sonipat was only 8%. Further, 14% of farmer households reported producing pulses, legumes, and nuts at home, while, in Sonipat, only 2% of farmer households were producing these crops. Milk production was more prevalent among farmer households in Sonipat (85%), while 43% of farmer households in Visakhapatnam were engaged into dairy farming i.e., milk production. Similarly, 20% of farmer households were producing oil and fats, whereas all farmer households in Visakhapatnam were dependent on the market.

### 3.2. Statistical Analyses

HDDS overall and seasonal: Distance to food markets (DFM) was the only factor that was significantly and negatively associated ($p < 0.05$) with farmer HDDS (overall, monsoon, winter, and summer) in Visakhapatnam (Table 4). Cropping intensity ($p < 0.1$) was positively associated with winter and summer HDDS in Visakhapatnam. In Sonipat, wealth index (WI) had a positive significant association ($p < 0.1$) with farmer HDDS (overall, monsoon, and winter), whereas cropping intensity was negatively associated ($p < 0.05$) with winter HDDS (Table 4).

**Table 4.** Regression results showing the agricultural and socioeconomic factors associated with farmer Household Dietary Diversity Score (HDDS overall, monsoon, winter, and summer) in Visakhapatnam and Sonipat with block fixed effects.

| Independent Factors | HDDS (Overall) | | HDDS (Monsoon) | | HDDS (Winter) | | HDDS (Summer) | |
|---|---|---|---|---|---|---|---|---|
| | Visakhapatnam | Sonipat | Visakhapatnam | Sonipat | Visakhapatnam | Sonipat | Visakhapatnam | Sonipat |
| Cropping Intensity (CI) | 0.059 | −0.072 | 0.046 | −0.079 | 0.088 * | −0.137 ** | 0.088 * | −0.054 |
| | −0.037 | −0.048 | −0.046 | −0.059 | −0.049 | −0.07 | −0.051 | −0.06 |
| Crop Diversity Index (CDI) | −0.029 | 0.0001 | −0.02 | 0.037 | −0.048 | −0.065 | −0.039 | 0.029 |
| | −0.027 | −0.051 | −0.033 | −0.062 | −0.035 | −0.074 | −0.036 | −0.064 |
| Family Education Index (FEI) | −0.014 | 0.0002 | −0.023 | −0.007 | −0.009 | −0.007 | −0.02 | 0.014 |
| | −0.033 | −0.041 | −0.04 | −0.05 | −0.043 | −0.059 | −0.045 | −0.051 |
| Income Diversity Index (IDI) | −0.004 | 0.022 | −0.026 | 0.025 | 0.001 | 0.043 | 0.01 | 0.015 |
| | −0.037 | −0.032 | −0.046 | −0.039 | −0.049 | −0.047 | −0.051 | −0.04 |
| Kitchen Garden (Y/N) | 0.094 | −0.078 | 0.116 | −0.068 | 0.088 | −0.094 | 0.15 | −0.128 |
| | −0.104 | −0.116 | −0.128 | −0.141 | −0.137 | −0.168 | −0.142 | −0.144 |
| Milk Production (Y/N) | 0.102 | 0.022 | 0.117 | −0.006 | 0.14 | 0.047 | 0.125 | 0.041 |
| | −0.068 | −0.063 | −0.084 | −0.077 | −0.09 | −0.092 | −0.093 | −0.079 |
| Distance to Food Markets (Kms) | −0.087 ** | −0.028 | −0.105 ** | −0.035 | −0.104 ** | −0.041 | −0.117 ** | −0.03 |
| | −0.037 | −0.033 | −0.046 | −0.04 | −0.049 | −0.047 | −0.051 | −0.041 |
| Wealth Index | −0.014 | 0.050 * | −0.025 | 0.064 * | −0.014 | 0.070 * | −0.014 | 0.053 |
| | −0.06 | −0.028 | −0.074 | −0.035 | −0.079 | −0.041 | −0.082 | −0.035 |
| Observations | 236 | 243 | 236 | 243 | 236 | 243 | 236 | 243 |
| $R^2$ | 0.135 | 0.036 | 0.107 | 0.03 | 0.135 | 0.058 | 0.139 | 0.024 |
| Adjusted $R^2$ | 0.101 | −0.001 | 0.071 | −0.007 | 0.1 | 0.021 | 0.105 | −0.014 |
| Residual Std. Error | 0.449 | 0.46 | 0.552 | 0.562 | 0.589 | 0.666 | 0.613 | 0.572 |
| Sample Size (*n*) | 227 | 234 | 227 | 234 | 227 | 234 | 227 | 234 |
| Block Fixed-Effects | Y | Y | Y | Y | Y | Y | Y | Y |
| Note: Significance codes | | | | | | * $p < 0.1$; ** $p < 0.05$ | | |

Note: All regressions were run with block-fixed effects as a robustness check.

### 3.3. Factor Importance

Using the relaimpo package in R, the relative importance of all agricultural and socioeconomic factors (Figure 4) associated with HDDS (overall, monsoon, winter, and summer) in Visakhapatnam and Sonipat was assessed. Distance to food markets (DFM) and cropping intensity (CI) emerged as the most important factor affecting household dietary diversity in Visakhapatnam. In Sonipat, wealth index (WI), and cropping intensity (CI) were the top two most important variables contributing to HDDS (overall and seasonal). The third most important factor contributing to HDDS (overall and seasonal) was crop diversity (CDI) in Visakhapatnam, while, in Sonipat, DFM and CDI were the third most important contributing factors to HDDS (overall and seasonal).

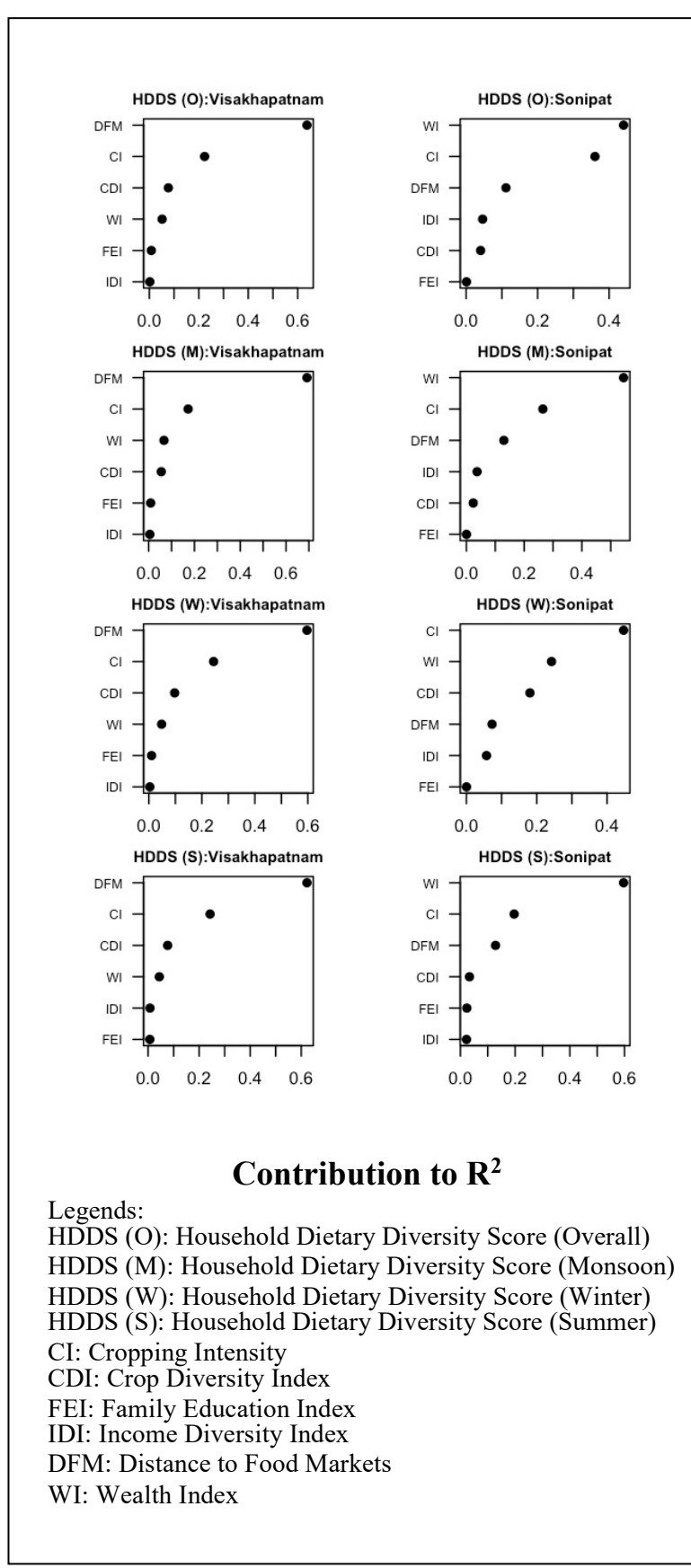

**Figure 4.** The relative importance of agricultural and socioeconomic factors associated with Household Dietary Diversity Score (HDDS) in different seasons in Visakhapatnam and Sonipat.

While looking at the relative contribution of these factors in quantitative terms, DFM emerged as the largest contributor (60–70%) to HDDS (overall and all seasons) in Visakhapatnam, while cropping intensity (i.e., growing more crops in a year) contributed 17–24% to HDDS (overall and all seasons) (Figure 4). In Sonipat, WI contributed 44%, 54%, and 60% to overall, monsoon, and summer HDDS, respectively, whereas CI was the largest contributor (45%) to the winter HDDS. CI was the second largest contributor (19–36%) to the overall, monsoon, and summer HDDS. DFM was the third largest contributor (11–12%) to overall, monsoon, and summer HDDS in Sonipat, while CDI's contribution to winter HDDS was 18%.

## 4. Discussion

Using primary data from 479 farmer households across 8 blocks and 24 villages of Visakhapatnam and Sonipat, this study investigated the associations between agricultural and socioeconomic factors and farmer household dietary diversity in India. This will help understand the implications of ongoing changes in cropping patterns in terms of cropping intensity and crop diversity, expanding farmer livelihood portfolios (i.e., a farmer looking for various allied and non-farm avenues for higher incomes), changing educational status of farmer households, and the integration of markets with rural households on farmer HDDS across three seasons—monsoon, winter, and summer.

Considering the educational status of an average farmer household, per capita education was significantly lower in Visakhapatnam than in Sonipat. The 2011 census (www.census2011.co.in, accessed on 20 July 2020) figures validate this variation as the rural literacy rate in Visakhapatnam was recorded as 54 with 63% among males and 45% among females, whereas the respective figures for Sonipat were much higher (Total—77%; male—86%; female— 66%). Regarding the economic status of farmer households, an average farmer household in Visakhapatnam earned USD 163 (Rs 12,918) from all income sources (2019–2020) whereas the respective figure for Sonipat was just USD 33 (Rs 2675) which is just 20% of the average figure reported by farmers in Visakhapatnam. However, as per the latest report by the NSSO [29], farmers' monthly income from all sources was USD 131 (Rs 10,480) and USD 287 (Rs 22,841) in Andhra Pradesh and Haryana, respectively, as compared with USD 128 (Rs 10,218) at the pan-India level. It seems that amidst the farmer protests at Delhi borders (near the study locations in Sonipat) at the time of the survey [30], farmers, particularly in Sonipat, underreported their income estimates, which may partly be an expression of their annoyance or frustration towards the central government or an effort to avoid any financial consequences of reporting higher farm incomes (note the central government's potential proposal to introduce income tax on agriculture, particularly for farmers with large holdings) [31]. A higher number of farmer households in Visakhapatnam were engaged in non-farming occupations compared to those in Sonipat, who were largely relying on crop production. The latest NSSO report validating these findings suggests that crop production contributed up to 26% to the farmers' household income in Visakhapatnam between July 2018 and June 2019, while the respective figure for Sonipat is 40% [29]; this means farmers in Visakhapatnam are exploring non-farm occupations such as casual labor, business, and dairy farming, whereas farmers in Sonipat are largely depending on crop production and dairy farming.

The average cropping intensity in Sonipat was significantly higher than in Visakhapatnam which could be due to the availability of water resources and agricultural advancements in Sonipat. Haryana and Punjab have been the hotspots of the green revolution since the mid-1960s [21,32]. Most farms in Visakhapatnam are dependent on rain while 96% of the land area in Sonipat is irrigated with one or the other source (www.crida.in, accessed on 25 July 2020; www.agricoop.nic.in, accessed on 25 July 2020). However, despite having fertile land and abundant water resources, farmers in Sonipat were concentrating on traditional crops such as wheat and rice. Further, although farmers in both study regions have access to land, water, and human resources, and are relatively educated, very few farmer households had a kitchen garden and one-fifth of farmer households reported

growing crops for domestic consumption without using fertilizers and pesticides. However, many studies in India and across the globe suggest that kitchen gardens play an important role, not only in improving the nutritional status of households, but also in yielding higher incomes to farmers [33–35]. Therefore, the concept of a kitchen garden could be considered a promising way to improve nutrition and livelihoods among farmer households in rural India.

The average HDDS (overall, monsoon, winter, and summer) in Visakhapatnam was significantly higher than that of Sonipat. Major food groups, such as cereals, root and tubers, vegetables, and fruits were consumed by all farmer households across all three seasons at both locations. However, none of the farmer households reported consuming non-vegetarian food groups in Sonipat, which is due to religious and cultural reasons. The NFHS report validates these findings and suggests that nearly 70% of the population in Haryana consumes vegetarian diets full of milk and curd [3]. While looking at the food sources, major food groups, such as cereals and milk, were largely produced by farmer households at home while, for the other food groups, they relied on close-by markets. On average, farmers had to travel 4-5 km to buy food items with a small variation in the two study locations.

While considering the regression results, cropping intensity, one of the major agricultural factors, was positively associated with farmer HDDS in the winter and summer seasons in Visakhapatnam, where cropping intensity is 129%. It suggests that increasing the cropping intensity may improve farmer HDDS in this region. However, in Sonipat, cropping intensity had a negative association with winter HDDS which needs further investigation. Although no paper suggested a direct association between cropping intensity and household dietary diversity, a couple of studies suggested that cropping intensity can expand the area under crops and enhance food security, particularly among smallholders into subsentence farming [36].

Distance to food markets had a significant and negative association with farmer HDDS (overall and across all three seasons) in Visakhapatnam, which suggests that the closer the markets, higher the farmer HDDS. However, a recent study [13] reported that distance to market was positively associated with individual dietary diversity scores among men, women, adolescents, and children in Haryana. Kumar et al. [14] endorsed these results, arguing that families wanting to consume diverse diets might need to travel farther distances to buy food items that were generally not available within their village/locality. However, our results seem more logical and support the integration of markets with rural households. Further, wealth index (WI) was positively associated with farmer HDDS (overall, monsoon, and winter), which simply suggests that the higher the family income, the higher the HDDS. Plenty of studies suggested that higher incomes are associated with improved household dietary diversity in India and other parts of the world [13,15,37].

Considering the relative importance of agricultural and socioeconomic factors associated with farmer HDDS (overall and seasonal), two factors, distance to food markets (60–70%) and cropping intensity (17–24%), emerged as the most important factors affecting household dietary diversity in Visakhapatnam. Although crop diversity did not have a significant association with farmer HDDS, it was the third most important factor contributing to HDDS (overall and seasonal) in Visakhapatnam. In Sonipat, wealth index (44–60%) and cropping intensity (45%) were the top two most important variables contributing to farmer HDDS (overall and seasonal). Distance to food markets (11–12%) and crop diversity (18%) were the third most important contributing factors to HDDS (overall and seasonal). A recent study reported distance to markets, crop diversity, and per-capita annual income as the major contributors to individual dietary diversity scores in Haryana [13].

## 5. Conclusions

Using primary data from 479 farmer households across 8 blocks and 24 villages of Visakhapatnam and Sonipat, this study examined the associations between agricultural and socioeconomic factors and farmer household dietary diversity to better understand the implications of ongoing changes in agricultural transitions in cropping patterns on farmer livelihood portfolios, the educational status of farmer households, and market integration in rural India. Regression results suggest that cropping intensity had a positive association with farmer HDDS in the winter and summer seasons which suggests that increasing cropping intensity may improve farmer HDDS in Visakhapatnam. Some studies indirectly endorse that increasing cropping intensity can help in the expansion of cultivable land area and improve food security, specifically for subsentence farmers. A socioeconomic factor, namely, distance to food markets, had a significant association with farmer HDDS, suggesting market integration can improve farmer HDDS across all seasons in Visakhapatnam. Another socioeconomic factor, wealth index, had a positive association with farmer HDDS in Sonipat, suggesting an income pathway to improve HDDS in rural Sonipat. Distance to food markets, cropping intensity, and crop diversity emerged as the three most important factors affecting farmer HDDS in Visakhapatnam. In Sonipat, wealth index, cropping intensity, and distance to food markets were the top three important agricultural and socioeconomic variables contributing to farmer HDDS across seasons.

In conclusion, our results suggest that improved market integration and intense and diversified cropping patterns can help enhance farmer HDDS in Visakhapatnam, whereas, in Sonipat, higher farmer incomes and market integration may improve farmer HDDS. Broadly speaking, the associations between agricultural and socioeconomic factors and farmer HDDS are complex; therefore, future policies that aim to improve household dietary diversity among farmer households in rural India would benefit by being targeted to a given location and context.

## 6. Limitations and Further Research Implications

This study has a few limitations. Firstly, some study results are not causal as they are correlational and based on observation data. However, we tried to better control for possible confounding effects that can occur at the regional scale by including block fixed effects in our regressions as a robustness check. Future work may collect panel data from the same farmers to understand further 'how changes in agricultural and socioeconomic variables within a given household influence changes in household dietary diversity.' Secondly, we collected household dietary intake data once a year using farmers' estimates for different seasons. Future studies may collect dietary information across different seasons in a year and investigate their associations with agricultural and socioeconomic factors individually. However, Rao and Raju [38] argue that diets in India do not vary much across seasons, which is also evident from our data (Tables 2 and 3). Finally, this study is a case study that compares only two regions, Visakhapatnam and Sonipat, which represent different agroclimatic, agricultural, and socioeconomic conditions within India. Our study outlines the potential implications for farmer household nutrition in two divergent regions of India; it does not attempt to extrapolate the results to all Indian states. Future research work should examine multiple states in India that represent a range of variations along a gradient to better attribute causal relationships between agricultural and socioeconomic factors and farmer HDDS across seasons.

**Author Contributions:** Methodology, S.S. and S.M.; Software, S.S.; Validation, S.S.; Formal analysis, S.S.; Investigation, S.S.; Resources, S.S., N.S.V., A.P.M., R.K. and P.J.; Data curation, S.S., N.S.V. and A.P.M.; Writing—original draft, S.S.; Writing—review & editing, S.S. and K.A.B.; Visualization, S.S.; Supervision, S.S., D.P. and S.M.; Project administration, D.P. and S.M.; Funding acquisition, D.P. and S.M. All authors have read and agreed to the published version of the manuscript.

**Funding:** This study was supported by The Sustainable and Healthy Food Systems (SHEFS) project, funded by the Wellcome Trust, UK (Grant number 205200/Z/16/Z) under the "Our Planet Our Health" programme.

**Institutional Review Board Statement:** The study was conducted in accordance with the Declaration of Helsinki, and approved by the Institutional Review Board (or Ethics Committee) of Centre for Chronic Disease Control (IRB00006330, 9 December 2020).

**Informed Consent Statement:** Informed consent was obtained from all subjects involved in the study.

**Data Availability Statement:** The datasets used and/or analyzed during the current study are available with CCDC, and can be obtained from Sailesh Mohan (co-author) on reasonable request.

**Acknowledgments:** We are thankful to the Wellcome Trust (UK) for providing funding and support to conduct this study. We are grateful to our SHEFS colleagues spread across UK, India, and Africa for their peer support in conceptualizing and designing this study. We are also thankful to Andrew Jones (University of Michigan, Ann Arbor, USA) for his valuable input at various stages of this project. We sincerely thank our data collection teams and their supervisors, who worked hard to collect data in very difficult situations. Last but not the least, we are thankful to the farmers and their families who spared long hours to participate in the survey.

**Conflicts of Interest:** The authors declare that they have no conflict of interest.

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
