# Peer review of "Agricultural and Socioeconomic Factors Associated with Farmer Household Dietary Diversity in India: A Comparative Study of Visakhapatnam and Sonipat"

_sustainability, doi:10.3390/su15042873_

Round 1

Reviewer 1 Report

The manuscript investigates the association of the socioeconomic and agricultural factors with household dietary diversity in two agro-climatically, socially, economically, and culturally divergent regions of India, Visakhapatnam and Sonipat. Limited studies address this aspect at the regional level.

The study is based on primary data from 479 farmer households.

The study deserves a linguistic revision. It presents inconsistent hyphenation and punctuation, article usage problems, missing a comma, and the verbs in the dependent clause are sometimes in the wrong tense. Moreover, the manuscript presents misplaced subordinate clause problems. Rereading the paper could solve typos issues, such as in lines 40-42.

The authors explain the data selection approach from a technical point of view, but the practical implications are unclear. Moreover, the selection based on FII indices is not justified.

The use of HDDS is not justified. Food security is a multidimensional dimension. Many indicators are available to capture these different aspects. The manuscript does not clarify the reason why HDDS was preferred.

Results are not discussed in light of the general literature.

Limitations to the study are well captured in the final session of the manuscript.

Author Response

Dear Sir/Madam 

I am extremely thankful for reviewing my document so carefully and providing very useful suggestions. I have revised the paper and made the suggested corrections.

Regarding FII, I have been using this sampling technique since 2009 and it is quite robust and published several papers using this sampling methodology. For this paper, I used FII to sample blocks and villages to collect data having enough variability within each location.

Regarding the use of HDDS, I honor your viewpoint. However, HDDS is a commonly used measurement for nutrition assessment. This paper is not focusing on food security.

Best regards

Reviewer 2 Report

Dear Authors,

in the manuscript submitted for review, a topic of great social and economic importance was taken up. The presented material shows the great contribution of the authors' work as well as great accuracy and diligence in the preparation of the materials. My only remark is that the Conclusions chapter could have been more factual, concise. Good luck with your further research!

Author Response

Dear Sir/Madam

I am extremely grateful for your review and very useful suggestions. I revised the paper incorporating reviewers' suggestions. Thanks again for great suggestions.

Best regards 

Reviewer 3 Report

The authors used primary data from 479 farmer households, we examined the associations between agricultural and socioeconomic factors and farmer household dietary diversity in Visakhapatnam and Sonipat, and they draw some valuable conclusions, which have certain reference value.However, the paper still has the following deficiencies.

1. In section of Discussion", this part lacks comparison and discussion with other scholars' research results.

2.Line 316. Usually the P value is less than 0.05, however, the P value in this study is less than 0.5.

Author Response

Dear Sir/Madam

Please accept my sincere thanks for providing a close review and noting down very essential and minute details. I have corrected the p-value in the text and in the table. I have revised some portions of the paper to incorporate reviewers' suggestions. Thanks again for providing very useful suggestions.

Best regards